# Mathematical Modeling of Capillary Drawing Stability for Hollow Optical Fibers

Vladimir Pervadchuk, Daria Vladimirova and Anna Derevyankina *

Department of Applied Mathematics, Perm National Research Polytechnic University, 614990 Perm, Russia
* Correspondence: al_derevyankina@mail.ru; Tel.: +7-919-700-92-40

**Abstract:** The stability problem solution of the manufacturing (drawing) of the quartz capillaries (pipes) for microstructured optical fibers (hole-assisted fiber) is important for determining the effective technological production modes. This importance is also caused by the high cost of fiber production and strict requirements for the accuracy of the fiber's geometric characteristics. Therefore, a theoretical approach to this problem is relevant and necessary. A modified capillary drawing model that takes into account inertial, viscous, and surface tension forces, as well as all types of heat transfer is proposed in the research. Within the framework of the linear theory of stability, a mathematical model of isothermal and nonisothermal capillary drawing has been developed. The stability of the process is studied depending on the drawing ratio and the Reynolds number. The analysis of the sensitivity of the process to perturbations in the boundary conditions is carried out. The secondary flow that occurs upon transition to the region of instability is also studied. It has been found that at draw ratios above critical values (instability region), undamped oscillations arise. The existence of optimal parameters of the heating element is shown: temperature distribution over the furnace surface and furnace radius, at which the stability of the process of drawing quartz tubes increases significantly (several times).

**Keywords:** photonic crystal fibers; capillary drawing; mathematical model; stability; calculation algorithms

## 1. Introduction

One of the modern and promising directions in the development of science-intensive industries is the production of optical fibers [1]. The areas of fiber technologies application are quite wide. These are fiber-optic lines for fast and high-quality communication, navigation, medicine, oil and gas industry, geology [2]. Today, much attention is paid to instrumentation which are using fiber-optic components [3]. Note that the process of manufacturing optical fibers is expensive and rather complicated [4]. At the same time, there is no complete study yet on the production problems. In some cases, this is due to the lack of information on these issues, and in some cases, due to the fact that the practical implementation of modern technologies often outstrips the process of full scientific description and research. Therefore, effective management of fiber production and increasing the stability of technological processes is certainly an urgent task. First of all, the above concerns special quartz optical fibers since they are of the greatest interest for use in various fields.

It is well-known that the full production cycle of optical fibers from raw materials to finished fiber can be represented as a number of independent production stages and four main stages can be distinguished among them: extrusion of quartz pipes from quartz raw materials; doping of quartz pipes; jacketing/scaling of quartz pipes; drawing fiber from the preform. The drawing process is the process of converting a quartz preform into fiber. It takes place in a special high-precision mechanical system called a drawing tower. A schematic of the drawing tower is shown in Figure 1. The tower feed mechanism (block B1) places the preform in the center of the precision furnace (B2), inside which the preform

melting process is started. The characteristic temperatures of the process are approximately 2000 °C (2273.15 K), close to the temperatures of the phase transition state of quartz, at which the viscosity of the substance changes sharply, creating the possibility of changing its shape [5]. When a preform with a diameter of 2–4 cm is transformed into a fiber with a diameter of only a few microns, an important target moment is the complete preservation of the relative geometric characteristics of quartz (in cross section). Following the exit from the furnace, the diameter of the hot quartz fiber is measured (B3). Then, as the fiber cools, it passes through an acrylate or silicone UV coating mechanism (B4) and is wound onto a coil (B5) [6].

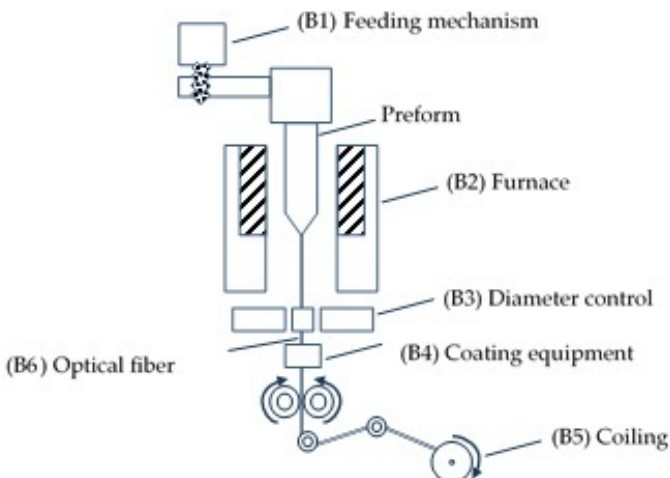

**Figure 1.** Scheme of the drawing tower.

It has been shown that the main optical and mechanical characteristics of quartz fibers depend on many factors [7]. First of all, this is the quality of the original preform. Here, an important role is played by the evaluation of its taper, the presence of small internal defects and screeds [8]. The selected technological modes of the production process are also important factors affecting the stability of the drawing process parameters. The critically important ones include the so-called drawing ratio (the ratio of the speeds of fiber drawing and quartz preform feeding), as well as the parameters of the furnace that heats the preform [9].

In addition, significant differences should be noted in the process of drawing solid and hollow (holey) fibers. In the first case, the control of the stability of the drawing process is reduced only to observe the geometric shape of the outer surface of the fiber. In the second case, it is also important to observe the inner surface in order to avoid collapse or swelling of the capillary [10].

Today, the problems of stability in the production of capillaries (tubes) are of particular interest, as far as capillaries are the main elements of photonic crystal fibers (PCF). A number of works [11–13] are devoted to mathematical modeling of technological processes of drawing capillaries, with an analysis of mathematical aspects and features of models, as well as numerical results of studies.

In [14,15], the regimes of drawing of hollow fibers in the non-thermal case are examined in detail and calculated, convective heat transfer with the environment is taken into account, but the radiant exchange has not been taken into account strictly. A.L. Yarin and P. Gospodinov played an important role in the study of the stability problems of single continuous fiber and capillaries. In their works [16,17], for the isothermal and non-thermal quasi-dimensional model of hoods of hollow fibers, resonance phenomena are studied. The influence of some parameters of the process on stability is also evaluated, but this applies only to isothermal case. In [18,19], the stability of the drawing process was studied from the point of view of maintaining the proportions of internal holes during the transition

from the workpiece to the fiber. The process of influence of surface tension forces in the transition zone has been studied in detail.

The general goal of this work is to systematize and solve the problem of the capillary drawing process stability, and, ultimately, to give recommendations on the choice of technological modes leading to a stable drawing process. In this paper, the formulating of the drawing mathematical model are based on the well-known [20] model, but the radiant heat transfer exchange in the energy equation is taken into account more accurately, and it is presented in an extended integral form. The particular attention is paid to the analysis of sensitivity to disturbances in the boundary conditions and secondary flows in the instability region. The influence of the parameters of the heating element on the stability is shown for the first time for a non-isothermal setting. New results have been obtained that make it possible to significantly increase the stability of capillary drawing under nonisothermal conditions. These results are important in the design of heating furnaces and in the selection of appropriate characteristics of the capillary drawing process.

## 2. Mathematical Models

### 2.1. Mathematical Model of Quartz Capillaries Drawing

Before proceeding directly to the stability analysis, it is necessary to consider the mathematical model of heat and mass transfer during drawing of quartz capillaries. The mathematical model of such process is described by a system of partial differential equations, namely, the equations of motion, continuity, and energy. It should be noted that the first works in this area were related to the drawing of solid fibers, and now there is a wide range of such models that allow us to consider problems in various formulations, as well as various types of heat transfer [21]. As for the mathematical models of capillary drawing (Figure 2), they were proposed later in the early 2000s. In 2000s, a study [20] presenting a quasi-one-dimensional model of capillary drawing was published.

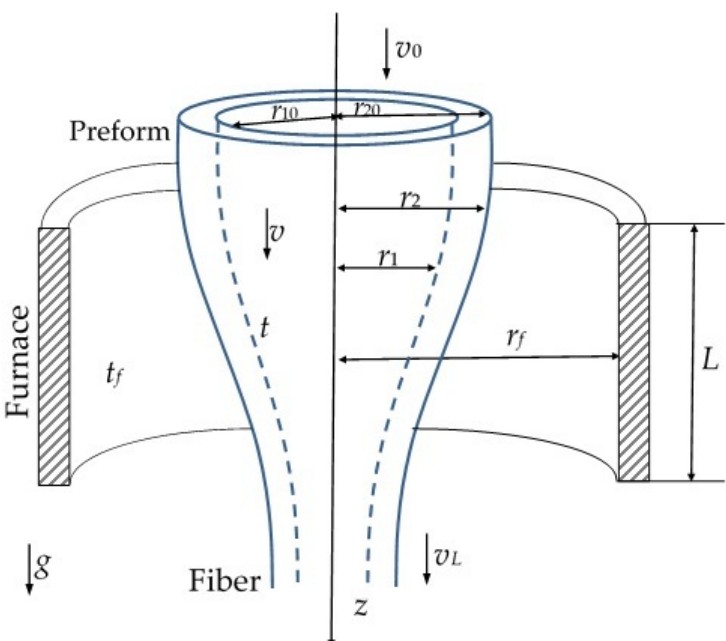

**Figure 2.** Configuration of the calculated area.

In this work, a modified pipe-drawing model was proposed. The model was obtained under the same assumptions as the model of the authors of the article [20]. These assumptions are as follows: (1) Flow and heat transfer are axisymmetric; (2) quartz melt is a viscous incompressible liquid; (3) the capillary wall thickness is small compared to its radius; (4) due to its smallness, the radial velocity is not taken into account; (5) changes in the axial velocity and temperature along the cross section of the capillary are small

compared to their changes along the *z* axis. The modified model, as well as the [20] model, takes into account inertial forces, viscous friction forces, and surface tension, as well as the combined action of all types of heat transfer: heat conduction, convection, and radiation. The modification of the model is caused by taking into account a radiant heat transfer more strictly.

In the proposed work, a modified model of tube drawing is proposed, which differs from the above model (radiant heat transfer is more strictly considered). Let us look at this aspect in more detail. When deriving the energy equation, considering radiant heat transfer, the laws of Planck, Stefan–Boltzmann, and Lambert are used [22]. It is known that the radiation spectrum of a blackbody with temperature *t* is determined by Planck's law:

$$\overline{E}_\nu = 2\pi \frac{\overline{h}\nu^3}{c^2} \cdot \frac{1}{\exp(\overline{h}\nu/k_B t) - 1} \, , \tag{1}$$

where *c* is the speed of light, $\lambda$ is the wavelength, $\nu$ is the frequency, at that $c = \lambda \cdot \nu$, $\overline{h}$ is Planck's constant, $k_B$ is Boltzmann constant.

For the integral flux density, according to the Stefan–Boltzmann law, we have ($\sigma_0$ is the Stefan–Boltzmann constant):

$$\overline{E} = \int_0^\infty \overline{E}_\nu d\nu = \sigma_0 t^4 \tag{2}$$

Radiant heat transfer between bodies is determined by the Lambert law: the amount of energy emitted by a surface element $dA_2$ in the direction of element $dA_1$ (Figure 3) is proportional to the energy emitted along the normal $E_n dA_2$ multiplied by the value of the elementary solid angle $d\Omega$ and $\cos\acute{\alpha}_2$, i.e.,

$$dQ = E_n dA_2 d\Omega \cos\alpha_2' \, , \quad E_n = \overline{E}/\pi \, . \tag{3}$$

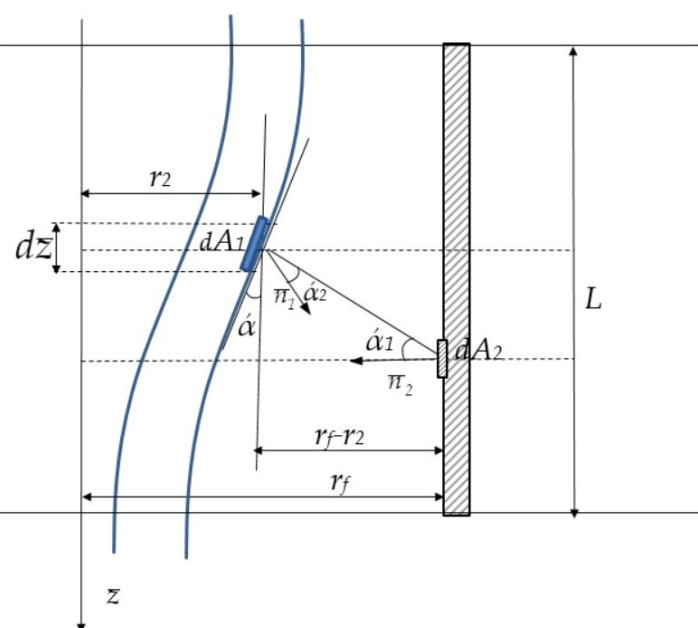

**Figure 3.** Radiant heat transfer during capillary drawing.

Since real materials do not completely absorb radiation, to consider this fact, the concept of emissivity $\varepsilon$, ($\varepsilon < 1$) is used and the laws of Planck and Stefan–Boltzmann are applied with a given correction factor $\varepsilon$. Let us consider $n_1$ as the normal to the point $A_1$ of the surface $\omega_1$ of the quartz fiber, and $A_1A_2$ as the segment connecting the points $A_1$ and

$A_2$. The resulting radiation flux at a surface point is the sum of the intrinsic radiation flux and the absorbed flux. Therefore, the following relation takes place, which expresses the law of energy conservation (in the blackbody approximation, $d\Omega = ds/r_2$):

$$q(A_1, \overline{\tau}) = \sigma_0 t^4(A_1, \overline{\tau}) - \frac{1}{\pi} \cdot \int_{\omega_1} \sigma_0 t^4(A_2, \overline{\tau}) \cdot \frac{1}{r^2} \cdot \cos(n_2, A_1 A_2) \cdot \cos(n_1, A_1 A_2) ds, \quad (4)$$

where integration is carried out over $\omega_1$ is a part of the thermoelement border, which is visible from point $A_1$.

Owing to the fact that heat flows for an elementary volume limited by sections $\overline{z} = z^*$, $\overline{z} = z^* + d\overline{z}$, and surface element $dA_1$, the energy equation was obtained. This equation was supplemented by the equations of continuity and motion [20,23]. As a result, the system of equations describing the motion and heat transfer during capillary drawing, taking into account the initial and boundary conditions, has taken the following form:

$$\frac{\partial r_1^2(\overline{z}, \overline{\tau})}{\partial \overline{\tau}} + \frac{\partial (v(\overline{z}, \overline{\tau}) r_1^2(\overline{z}, \overline{\tau}))}{\partial \overline{z}} = \frac{P_0 r_1^2(\overline{z}, \overline{\tau}) r_2^2(\overline{z}, \overline{\tau}) - \widetilde{\gamma} r_1(\overline{z}, \overline{\tau}) r_2(\overline{z}, \overline{\tau})(r_1(\overline{z}, \overline{\tau}) + r_2(\overline{z}, \overline{\tau}))}{\overline{\mu}(t(\overline{z}, \overline{\tau})) \cdot (r_2^2(\overline{z}, \overline{\tau}) - r_1^2(\overline{z}, \overline{\tau}))},$$

$$\frac{\partial r_2^2(\overline{z}, \overline{\tau})}{\partial \overline{\tau}} + \frac{\partial (v(\overline{z}, \overline{\tau}) r_2^2(\overline{z}, \overline{\tau}))}{\partial \overline{z}} = \frac{P_0 r_1^2(\overline{z}, \overline{\tau}) r_2^2(\overline{z}, \overline{\tau}) - \widetilde{\gamma} r_1(\overline{z}, \overline{\tau}) r_2(\overline{z}, \overline{\tau})(r_1(\overline{z}, \overline{\tau}) + r_2(\overline{z}, \overline{\tau}))}{\overline{\mu}(t(\overline{z}, \overline{\tau}))(r_2^2(\overline{z}, \overline{\tau}) - r_1^2(\overline{z}, \overline{\tau}))},$$

$$\rho(r_2^2(\overline{z}, \overline{\tau}) - r_1^2(\overline{z}, \overline{\tau})) \left( \frac{\partial v(\overline{z}, \overline{\tau})}{\partial \overline{\tau}} + v(\overline{z}, \overline{\tau}) \frac{\partial v(\overline{z}, \overline{\tau})}{\partial \overline{z}} - g \right) = \frac{\partial}{\partial \overline{z}} \left( 3\overline{\mu}(t(\overline{z}, \overline{\tau})) \cdot (r_2^2(\overline{z}, \overline{\tau}) - r_1^2(\overline{z}, \overline{\tau})) \frac{\partial v(\overline{z}, \overline{\tau})}{\partial \overline{z}} + \widetilde{\gamma}(r_2(\overline{z}, \overline{\tau}) + r_1(\overline{z}, \overline{\tau})) \right),$$

$$(r_2^2(\overline{z}, \overline{\tau}) - r_1^2(\overline{z}, \overline{\tau})) \left( \frac{\partial t(\overline{z}, \overline{\tau})}{\partial \overline{\tau}} + v(\overline{z}, \overline{\tau}) \frac{\partial t(\overline{z}, \overline{\tau})}{\partial \overline{z}} \right) \rho C_p = \frac{\partial}{\partial \overline{z}} \left( \lambda_{ef}(r_2^2(\overline{z}, \overline{\tau}) - r_1^2(\overline{z}, \overline{\tau})) \frac{\partial t(\overline{\overline{z}}, \overline{\tau})}{\partial \overline{z}} \right)$$

$$-2r_1(\overline{z}, \overline{\tau}) \sqrt{1 + r_1'^2(\overline{z}, \overline{\tau})} \cdot \overline{\alpha}_1 \cdot (t(\overline{z}, \overline{\tau}) - t_{in}) - 2r_2(\overline{z}, \overline{\tau}) \sqrt{1 + r_2'^2(\overline{z}, \overline{\tau})} \cdot [\omega_2 \varepsilon \cdot n_c^2 \sigma_0 \cdot (t^4(\overline{z}, \overline{\tau}) - t_{out}^4) + \overline{\alpha}_2(t(\overline{z}, \overline{\tau}) - T_a)]$$

$$+4n_c^2 \sigma_0 \cdot r_2(\overline{z}, \overline{\tau}) \cdot r_f \cdot (r_f - r_2(\overline{z}, \overline{\tau})) \cdot \int_0^L \frac{\left(\overline{\beta} \varepsilon_f t_f^4(\eta, \overline{\tau}) - \varepsilon \cdot t^4(\overline{z}, \overline{\tau})\right)\left((r_f - r_2(\overline{z}, \overline{\tau})) + |r_2'(\overline{z}, \overline{\tau})|(\overline{z} - \eta)\right)}{\left((\eta - \overline{z})^2 + (r_f - r_2(\overline{z}, \overline{\tau}))^2\right)^2} d\eta,$$

$$v(\overline{z}, 0) = v_s(\overline{z}), \ v(0, \overline{\tau}) = v_0, \ v(L, \overline{\tau}) = v_L,$$

$$r_1(\overline{z}, 0) = r_{1s}(\overline{z}), \ r_1(0, \overline{\tau}) = r_{10}, \ r_2(\overline{z}, 0) = r_{2s}(\overline{z}), \ r_2(0, \overline{\tau}) = r_{20},$$

$$t(\overline{z}, 0) = t_s(\overline{z}), \ t(0, \overline{\tau}) = t_0, \ \left. \frac{\partial t}{\partial z} \right|_{z=L} = 0. \tag{5}$$

Here $v_s(\overline{z})$ is the initial speed, [m/s]; $v_0$ is the fiber feed rate, [m/s]; $v_L$ is the fiber drawing speed, [m/s]; $r_{1s}(\overline{z})$, $r_{2s}(\overline{z})$ are initial values of the inner and outer radius of the capillary, [m]; $r_{10}$, $r_{20}$ are the inner and outer radius of the preform, [m]; $t_s(\overline{z})$ is the initial temperature, [°C]; $t_0$ is the preform temperature, [°C].

The list of symbols used and their units of measurement are given in Table 1.

**Table 1.** Symbols used and their units of measurement.

| Symbols | Description | Symbols | Description |
|---|---|---|---|
| $\overline{z}$ | Longitudinal coordinate, [m] | $t_{in}$ | Gas temperature inside the tube, [°C] |
| $\overline{\tau}$ | Time, [s] | $P_0$ | Difference between internal and external pressure, [Pa] |
| $r_2(\overline{z}, \overline{\tau})$ | Outer radius of capillary, [m] | $C_p$ | Melt thermal conductivity, [J/g°C] |
| $r_1(\overline{z}, \overline{\tau})$ | Inner radius of capillary, [m] | $\rho$ | Melt density, [g·m$^3$] |
| $v(\overline{z}, \overline{\tau})$ | Melt flow rate, [m/s] | $\overline{\beta}$ | Reflection coefficient, [1] |
| $t(\overline{z}, \overline{\tau})$ | Melt temperature, [°C] | $\varepsilon_f$ | Degree of the heating element emissivity, [1] |
| $\overline{\mu}(t(\overline{z}, \overline{\tau}))$ | Viscosity of the quartz melt, [Pa·s] | $\varepsilon$ | Emissivity of the quartz melt, [1] |
| $t_f(\overline{z}, \overline{\tau})$ | Furnace temperature, [°C] | $\widetilde{\gamma}$ | Surface tension coefficient, [N/m] |
| $L$ | Heating zone length, [m] | $\overline{\alpha}_1$ | Heat transfer coefficient from the inner surface of the furnace, [W/(m$^2$·°C)] |
| $t_{out}$ | Gas temperature outside the tube, [°C] | $\overline{\alpha}_2$ | Heat transfer coefficient from the outer surface of the furnace, [W/(m$^2$ °C)] |
| $T_a$ | Ambient temperature, [°C] | $\lambda_t$ | Melt molecular thermal conductivity, [W/(m$^2$ °C)] |

**Table 1.** *Cont.*

| Symbols | Description | Symbols | Description |
|---|---|---|---|
| $r_f$ | Furnace radius, [m] | $\lambda_{ef}$ | Effective coefficient of thermal conductivity (molecular and radiative), [1] |
| $\omega_2$ | Preform surface emissivity coefficient outside of furnace, [1] | $n_c$ | Refractive index of gas, [1] |
| $\sigma_0$ | Stefan-Boltzmann constant, [1] | | |

To check the adequacy of the modified mathematical model (5), the numerical calculations were compared with the experimental data given in [23]. The cited work presents the results of 24 measurements of the finished product radii during capillary drawing. The preform was made of Suprasil F300 glass with an outer radius $r_{10} = 14 \times 10^{-3}$ m and an inner radius $r_{20} = 12 \times 10^{-3}$ m. The preform was fed into the furnace at a constant speed, and the upper end of the tube was left open to the atmosphere. Feed speed $v_0$ varied from $2 \times 10^{-3}$ m/min to $8 \times 10^{-3}$ m/min, drawing speed $v_L$ varied from 0.6 m/min to 1.2 m/min at furnace temperatures of 1900 °C, 1950 °C, and 2000 °C. During the experiment, both external and internal diameters were measured.

Numerical studies of the movement and heat transfer during fiber drawing hereinafter were made in the Comsol Multiphysics 5.2 (finite element method for PDE) environment, the results for the inner diameter are shown in Figure 4. As can be seen from the figure, the numerical results obtained in the work coincided quite well with the experimental data. A similar picture was also observed for the outer diameter.

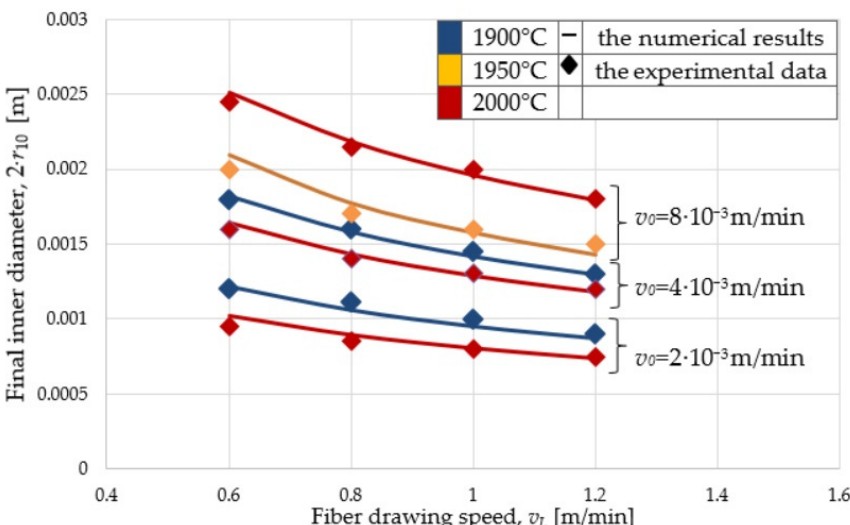

**Figure 4.** Dependence of the inner diameter of the capillary on the feed and draw speed, furnace temperature.

Thereby, the modified model (5) proposed in the paper describes the process of drawing a quartz tube with a sufficiently high accuracy, considering all types of heat transfer occurring inside the furnace.

Subsequently, for the convenience of generalizing the results obtained in the course of a numerical study of stability, the equations of the system were written in a dimensionless form. At the same time, it was assumed that

$$z = \frac{\bar{z}}{L}, \ \tau = \frac{\bar{\tau} v_L}{L}, \ V(z, \tau) = \frac{v(\bar{z}, \bar{\tau})}{v_L}, \ R_1(z, \tau) = \frac{r_1(\bar{z}, \bar{\tau})}{L}, \ R_2(z, \tau) = \frac{r_2(\bar{z}, \bar{\tau})}{L}, \ T(z, \tau) = \frac{t(\bar{z}, \bar{\tau})}{T_a},$$

$$\lambda = \frac{\lambda_{ef}}{\lambda_t}, \ \mu = \frac{\bar{\mu}}{\mu_0}, \ R_f = \frac{r_f}{L}, \ T_{in} = \frac{t_{in}}{T_a}, \ T_{out} = \frac{t_{out}}{T_a}, \ T_f(z, \tau) = \frac{t_f(\bar{z}, \bar{\tau})}{T_a}.$$

As a result, system (5) took the following form:

$$\left(R_2{}^2(z,\tau) - R_1{}^2(z,\tau)\right)\left(\frac{\partial V(z,\tau)}{\partial \tau} + V(z,\tau)\frac{\partial V(z,\tau)}{\partial z}\right)$$

$$= \frac{3}{Re}\cdot\frac{\partial}{\partial z}\left(\left(R_2{}^2(z,\tau) - R_1{}^2(z,\tau)\right)\cdot\mu(T(z,\tau))\cdot\frac{\partial V(z,\tau)}{\partial z}\right) + \frac{(R_2{}^2(z,\tau)-R_1{}^2(z,\tau))}{Fr} + \frac{1}{We}\cdot\frac{\partial(R_1(z,\tau)+R_2(z,\tau))}{\partial z},$$

$$\frac{\partial R_1{}^2(z,\tau)}{\partial \tau} + \frac{\partial(V(z,\tau)R_1{}^2(z,\tau))}{\partial z} = \frac{La\,R_1{}^2(z,\tau)R_2{}^2(z,\tau)-\frac{1}{Ma}R_1(z,\tau)R_2(z,\tau)(R_1(z,\tau)+R_2(z,\tau))}{\mu(T(z,\tau))\cdot(R_2{}^2(z,\tau)-R_1{}^2(z,\tau))},$$

$$\frac{\partial R_2{}^2(z,\tau)}{\partial \tau} + \frac{\partial(V(z,\tau)R_2{}^2(z,\tau))}{\partial z} = \frac{La\,R_1{}^2(z,\tau)R_2{}^2(z,\tau)-\frac{1}{Ma}R_1(z,\tau)R_2(z,\tau)(R_1(z,\tau)+R_2(z,\tau))}{\mu(T(z,\tau))\cdot(R_2{}^2(z,\tau)-R_1{}^2(z,\tau))},$$

$$\left(R_2{}^2(z,\tau) - R_1{}^2(z,\tau)\right)\left(\frac{\partial T(z,\tau)}{\partial \tau} + V(z,\tau)\frac{\partial T(z,\tau)}{\partial z}\right) = \frac{1}{Pe}\cdot\frac{\partial}{\partial z}\left(\lambda\left(R_2{}^2(z,\tau) - R_1{}^2(z,\tau)\right)\frac{\partial T(z,\tau)}{\partial z}\right)$$

$$-2R_1(z,\tau)\sqrt{1+R_1'{}^2(z,\tau)}\cdot St_1\cdot(T(z,\tau)-T_{in}) - 2R_2(z,\tau)\sqrt{1+R_2'{}^2(z,\tau)}\cdot St_2\cdot(T(z,\tau)-1)$$

$$-2\chi_1 R_2(z,\tau)\sqrt{1+R_2'{}^2(z,\tau)}\cdot(T^4(z,\tau)-T_{out}{}^4) \tag{6}$$

$$+4\chi_2 R_2(z,\tau)\cdot R_f\cdot\left(R_f - R_2(z,\tau)\right)\cdot\int\limits_0^1 \frac{\left(\bar\beta\varepsilon_f T_f^4(\eta,\tau)-\varepsilon T^4(z,\tau)\right)\left((R_f-R_2(z,\tau))+\left|R_2'(z,\tau)\right|\cdot(z-\eta)\right)}{\left((\eta-z)^2+(R_f-R_2(z,\tau))^2\right)^2}\,d\eta$$

$$V(z,0) = \frac{v_s(z)}{v_L} = V_s(z),\quad V(0,\tau) = \frac{v_0}{v_L} = \frac{1}{E},\quad V(1,\tau) = 1,$$

$$R_1(z,0) = \frac{r_{1\,s}(z)}{L} = R_{1\,s}(z),\quad R_1(0,\tau) = \frac{r_{10}}{L} = R_{10},$$

$$R_2(z,0) = \frac{r_{2\,S}(z)}{L} = R_{2s}(z),\quad R_2(0,\tau) = \frac{r_{20}}{L} = R_{20},$$

$$T(z,0) = \frac{t_s(z)}{T_a} = T_s(z),\quad T(0,\tau) = \frac{t_0}{T_a},\quad \frac{\partial T}{\partial z}\bigg|_{z=1} = 0.$$

The dimensionless quantities are listed in Table 2.

**Table 2.** Dimensionless quantities.

| Symbols | Description | Symbols | Description |
|---|---|---|---|
| $Re = \dfrac{\rho v_L L}{\mu_0}$ | Reynolds number | $La = \dfrac{P_0 L}{\mu_0 v_L}$ | Criterion for the interaction of capillary forces |
| $Fr = \dfrac{v_L^2}{Lg}$ | Froude number | $Ma = \dfrac{\mu_0 v_L}{\widetilde{\gamma}}$ | Criterion for the interaction of forces of molecular friction |
| $We = \dfrac{\rho L v_L^2}{\widetilde{\gamma}}$ | Weber number | $Pe = \dfrac{\rho C_p v_L L}{\lambda_t}$ | Peclet number |
| $\chi_1 = \dfrac{\omega_2\varepsilon\cdot n_c^2\sigma_0\overline{T}_a^3}{\rho C_p v_L}$ | Dimensionless complexes 1, 2 | $St_1 = \dfrac{\overline{\alpha}_1}{\rho C_p v_L}$ | Stanton's criterion |
| $\chi_2 = \dfrac{n_c^2\sigma_0\overline{T}_a^3}{\rho C_p v_L}$ | | $St_2 = \dfrac{\overline{\alpha}_2}{\rho C_p v_L}$ | |

Subsequently, the analysis of the process stability was made on the basis of the system equations (6).

## 2.2. Mathematical Model of the Non-Isothermal Process Stability of the Quartz Capillary Drawing

The property of smallness of the perturbation is essential in the definition of stability, and this property can serve as a basis for neglecting the products of perturbations of the desired functions in the process of studying stability. In other words, the analysis of a system linearized in the vicinity of its stationary state can replace the analysis of the original nonlinear system [24–26].

Accordingly, at the first stage, linearization was performed, in which the parameters determining the state of the system were divided into main $\overline{F}(z)$ and perturbing ones $\widetilde{F}(z,\tau)$:

$$F(z, \tau) = \overline{F}(z) \cdot (1 + \widetilde{F}(z, \tau)),$$
$$F(z, \tau) \in (V(z, \tau), R_1(z, \tau), R_2(z, \tau), T(z, \tau)),$$
$$\overline{F}(z) \in (\overline{V}(z), \overline{R}_1(z), \overline{R}_2(z), \overline{T}(z)),$$
$$\widetilde{F}(z, \tau) \in (\widetilde{V}(z, \tau), \widetilde{R}_1(z, \tau), \widetilde{R}_2(z, \tau), \widetilde{T}(z, \tau))$$

The main parameters are stationary solutions of system (3). As a result, a system was obtained that describes the evolution of perturbing influences. To simplify the notation, everywhere below $\widetilde{F}(z, \tau) = \widetilde{F}$, $\overline{F}(z) = \overline{F}$.

With this remark in mind, the linearized system of equations takes the form

$$\frac{\partial \widetilde{V}}{\partial \tau} = \frac{3}{Re} \cdot \frac{\partial^2 \widetilde{V}}{\partial z^2} + \beta_1(z)\frac{\partial \widetilde{V}}{\partial z} + \beta_2(z)\widetilde{V} + \alpha_1(z)\frac{\partial \widetilde{R}_2}{\partial z} + \alpha_2(z)\widetilde{R}_2 + \theta_1(z)\frac{\partial \widetilde{R}_1}{\partial z} + \theta_2(z)\widetilde{R}_1 + \varphi_1(z)\frac{\partial \widetilde{T}}{\partial z} + \varphi_2(z)\widetilde{T},$$

$$-\frac{\partial \widetilde{R}_1}{\partial \tau} = \beta_3(z)\frac{\partial \widetilde{V}}{\partial z} + \beta_4(z)\widetilde{V} + \theta_3(z)\frac{\partial \widetilde{R}_1}{\partial z} + \theta_4(z)\widetilde{R}_1 + \alpha_3(z)\widetilde{R}_2,$$

$$-\frac{\partial \widetilde{R}_2}{\partial \tau} = \beta_3(z)\frac{\partial \widetilde{V}}{\partial z} + \beta_5(z)\widetilde{V} + \theta_3(z)\frac{\partial \widetilde{R}_1}{\partial z} + \theta_5(z)\widetilde{R}_1 + \alpha_4(z)\widetilde{R}_2,$$

$$\frac{\partial \widetilde{T}}{\partial \tau} = \frac{\lambda}{Pe} \cdot \frac{\partial^2 \widetilde{T}}{\partial z^2} + \phi_3(z)\frac{\partial \widetilde{T}}{\partial z} + \varphi_4(z)\widetilde{T} + \alpha_5(z)\frac{\partial \widetilde{R}_2}{\partial z} + \alpha_6(z)\widetilde{R}_2 + \theta_6(x)\frac{\partial \widetilde{R}_1}{\partial z} + \theta_7(z)\widetilde{R}_1 + \beta_6(z)\widetilde{V},$$

$$\widetilde{V}(z,0) = 0, \ \widetilde{V}(0,\tau) = 0, \ \widetilde{V}(1,\tau) = 0,$$

$$\widetilde{R}_1(z,0) = 0, \ \widetilde{R}_1(0,\tau) = 0, \ \widetilde{R}_2(z,0) = 0, \ \widetilde{R}_2(0,\tau) = 0,$$

$$\widetilde{T}(z,0) = 0, \ \widetilde{T}(0,\tau) = 0, \ \frac{\partial \widetilde{T}}{\partial z}\Big|_{z=1} = 0.$$

(7)

The corresponding coefficients of system (7) depend only on the stationary solution of the original nonlinear system (6) and have the form

$$\alpha_1(z) = \frac{6\mu \overline{V}' \overline{R}_2^2}{\overline{R}^2 \overline{V} Re} + \frac{\overline{R}_2}{We \overline{V} \overline{R}^2}, \alpha_2(z) = \frac{-2 \overline{R}_2^2 \overline{V}}{\overline{R}^2} + \frac{6}{\overline{R}^2 \overline{V} Re} \cdot \frac{d}{dz}\left(\mu \overline{R}_2^2 \frac{d\overline{V}}{dz}\right) + \frac{2 \overline{R}_2^2}{\overline{V} Fr \overline{R}^2} + \frac{\overline{R}_2'}{We \overline{V} \overline{R}^2},$$

$$\alpha_3(z) = -\frac{\overline{R}_2}{2 \overline{R}_1^2}\left(\frac{2 La \overline{R}_1^2 \overline{R}_2 - \frac{2}{Ma}\overline{R}_1 \overline{R}_2 - \frac{1}{Ma}\overline{R}_1^2}{\mu \overline{R}^2}\right) + \frac{\overline{R}_2}{2 \overline{R}_1^2}\left(\frac{2 \overline{R}_2(La \overline{R}_1^2 \overline{R}_2^2 - \frac{1}{Ma}\overline{R}_1 \overline{R}_2(\overline{R}_1 + \overline{R}_2))}{\mu(\overline{R}_1^2 - \overline{R}_2^2)^2}\right),$$

$$\alpha_4(z) = \frac{1}{\overline{R}_2^2} \cdot \frac{d}{dz}\left(\overline{R}_2^2 \overline{V}\right) - \frac{1}{2 \overline{R}_2}\left(\frac{2 La \overline{R}_1^2 \overline{R}_2 - \frac{2}{Ma}\overline{R}_1 \overline{R}_2 - \frac{1}{Ma}\overline{R}_1^2}{\mu \overline{R}^2} - \frac{2 \overline{R}_2(La \overline{R}_1^2 \overline{R}_2^2 - \frac{1}{Ma}\overline{R}_1 \overline{R}_2(\overline{R}_1 + \overline{R}_2))}{\mu(\overline{R}_1^2 - \overline{R}_2^2)^2}\right),$$

$$\alpha_5(z) = \frac{1}{\overline{R}^2 \overline{T}} \cdot \left(\frac{2}{Pe}\frac{d}{dz}\left(\lambda \overline{R}_2^2 \frac{d\overline{T}}{dz}\right) - 2\overline{R}'_2 \overline{R}_2^2 St_2(\overline{T} - 1) - 2\overline{R}'_2 \overline{R}_2^2 (\overline{T}^4 - \overline{T}_{out}^4)\chi_1\right.$$

$$\left. + 4\chi_2 R_f \overline{R}_2^2\left(R_f - \overline{R}_2\right)k \cdot \int_0^1 \frac{(\beta \varepsilon_f T_f^4 - \varepsilon \overline{T}^4)(z - \eta)}{\left[(\eta - z)^2 + (R_f - \overline{R})^2\right]^2}d\eta\right),$$

$$\alpha_6(z) = \frac{1}{\overline{R}^2 \overline{T}} \cdot \left(\frac{2}{Pe}\frac{d}{dz}\left(\lambda \overline{R}_2^2 \frac{d\overline{T}}{dz}\right) - 2\overline{R}_2 \overline{T}(1 + \frac{3}{2}\overline{R}'_2{}^2) St_2(\overline{T} - 1) - 2\overline{R}_2(\overline{T}^4 - \overline{T}_{out}^4)(1 + \frac{3}{2}\overline{R}'_2{}^2)\chi_1 - 2\overline{V}\overline{R}_2^2 \frac{d\overline{T}}{dz}\right.$$

$$+ 4\chi_2 R_p \overline{R}_2 \int_0^1 (\beta \varepsilon_f T_f^4 - \varepsilon \overline{T}^4)\frac{(R_f - \overline{R}_2)(R_f - 3\overline{R}_2) + |\overline{R}'_2|(z - \eta)(R_f - 2\overline{R}_2)}{\left[(\eta - z)^2 + (R_f - \overline{R}_2)^2\right]^2}$$

$$\left. + \frac{(4\overline{R}_2)(R_f - \overline{R}_2)(R_f - \overline{R}_2 + |\overline{R}'_2|(z - \eta))}{\left[(\eta - z)^2 + (R_f - \overline{R}_2)^2\right]^3}d\eta\right),$$

$$\beta_1(z) = \frac{3}{\overline{R}^2 \overline{V} Re}\left((\mu \overline{R}^2 \overline{V})' + (\mu \overline{R}^2 \overline{V}')\right) - \overline{V}, \beta_2(z) = \frac{3}{\overline{R}^2 \overline{V} Re}\frac{d}{dz}(\mu \overline{R}^2 \overline{V}') - 2\overline{V}', \beta_3(z) = \frac{\overline{V}}{2}, \beta_4(z) = \frac{1}{2 \overline{R}_1^2}\frac{d}{dz}(\overline{R}_1^2 \overline{V}),$$

$$\beta_5(z) = \frac{1}{2 \overline{R}_2^2}\frac{d}{dz}(\overline{R}_2^2 \overline{V}), \beta_6(z) = -\frac{\overline{V}\overline{T}'}{\overline{T}}, \theta_1(z) = \frac{-6 \overline{R}_1^2 \overline{V}' \mu}{\overline{R}^2 \overline{V} Re} + \frac{\overline{R}_1}{We \overline{V} \overline{R}^2},$$

$$\theta_2(z) = \frac{-6}{\overline{R}^2 \overline{V} Re} \cdot \frac{d}{dz}\left(\mu \overline{R}_1^2 \frac{d\overline{V}}{dz}\right) + \frac{2\overline{V}' \overline{R}_1^2}{\overline{R}^2} - \frac{2 \overline{R}_1^2}{\overline{V} Fr \overline{R}^2} + \frac{\overline{R}_1'}{We \overline{V} \overline{R}^2}, \theta_3(z) = \overline{V},$$

$$\theta_4(z) = \frac{1}{\overline{R}_1^2} \cdot \frac{d}{dz}\left(\overline{R}_1^2 \overline{V}\right) - \frac{1}{2 \overline{R}_1}\left(\frac{2 La \overline{R}_1 \overline{R}_2^2 - \frac{2}{Ma}\overline{R}_1 \overline{R}_2 - \frac{1}{Ma}\overline{R}_2^2}{\mu \overline{R}^2} - \frac{2 \overline{R}_1(La \overline{R}_1^2 \overline{R}_2^2 - \frac{1}{Ma}\overline{R}_1 \overline{R}_2(\overline{R}_1 + \overline{R}_2))}{\mu(\overline{R}_1^2 - \overline{R}_2^2)^2}\right),$$

$$\theta_5(z) = -\frac{\overline{R}_1}{2\,\overline{R}_2{}^2}\left(\frac{2\,La\,\overline{R}_1\,\overline{R}_2{}^2 - \frac{2}{Ma}\overline{R}_1\,\overline{R}_2 - \frac{1}{Ma}\overline{R}_2{}^2}{\mu\overline{R}^2}\right) - \frac{\overline{R}_1}{2\,\overline{R}_2{}^2}\left(\frac{2\,\overline{R}_1\left(La\,\overline{R}_1{}^2\,\overline{R}_2{}^2 - \frac{1}{Ma}\overline{R}_1\,\overline{R}_2(\overline{R}_1 + \overline{R}_2)\right)}{\mu(\overline{R}_1{}^2 - \overline{R}_2{}^2)^2}\right),$$

$$\theta_6(z) = \frac{1}{\overline{R}^2\overline{T}}\cdot\left[-\frac{2}{Pe}\left(\lambda\overline{R}_1{}^2\frac{d\overline{T}}{dz}\right) - 2St_1\overline{R}_1{}^2\overline{R}_1'(\overline{T} - \overline{T}_{in})\right]$$

$$\theta_7(z) = \frac{1}{\overline{R}^2\overline{T}}\cdot\left[-\frac{2}{Pe}\frac{d}{dz}\left(\lambda\overline{R}_1{}^2\frac{d\overline{T}}{dz}\right) - 2St_1\overline{R}_1(\overline{T} - \overline{T}_{in})(1 + \tfrac{3}{2}\overline{R}_1'^2) + 2\overline{V}\overline{R}_1{}^2\frac{d\overline{T}}{dz}\right]$$

$$\varphi_1(z) = -\frac{3a_2\mu\,\overline{T}\overline{V}'}{Re\overline{V}}, \varphi_2(z) = -\frac{1}{Re\overline{R}^2\overline{V}}\cdot\frac{d}{dz}\left(3\mu\,a_2\overline{T}\overline{R}^2\frac{d\overline{V}}{dz}\right), \varphi_3(z) = \frac{\left(\lambda\,\overline{R}^2\overline{T}\right)' + \lambda\overline{R}^2\overline{T}'}{\overline{R}^2\overline{T}\,Pe} - \overline{V},$$

$$\varphi_4(z) = \frac{1}{\overline{R}^2\overline{T}}\cdot\left(\frac{1}{Pe}\frac{d}{dz}\left(\lambda\overline{R}^2\frac{d\overline{T}}{dz}\right) - 2\overline{R}_1\overline{T}\sqrt{1 + \overline{R}_1'^2}\,St_1 - 2\overline{R}_1\overline{T}\sqrt{1 + \overline{R}_1'^2}\,St_1 - 2\overline{R}_2\overline{T}\sqrt{1 + \overline{R}_2'^2}\,St_2\right.$$

$$\left. -8\overline{R}_2\overline{T}^4\sqrt{1 + \overline{R}_2'^2}\,\chi_1 - \overline{V}\overline{R}^2\frac{d\overline{T}}{dz} - 16\chi R_f\varepsilon(R_f - \overline{R}_2)\overline{T}^4\cdot\int\limits_0^1\frac{R_f - \overline{R}_2 + \left|\overline{R}_2'\right|(z - \eta)}{\left[(\eta - z)^2 + (R_f - \overline{R}_2)^2\right]^2}d\eta\right),$$

here: $\overline{R}^2 = \overline{R}^2{}_1 - \overline{R}^2{}_2$.

At the next stage, the method of separation of variables was applied, according to which the unknowns in the linearized equations of system (7) were represented as follows: $\widetilde{F}(z,\tau) = \widetilde{f}(z)\cdot e^{-i\omega\tau}$, $\widetilde{f}(z) \in (\widetilde{v}(z), \widetilde{t}(z), \widetilde{r}_1(z), \widetilde{r}_2(z))$.

After separation of variables, system (7) was reduced to a system of linear ordinary differential equations:

$$\frac{3\mu}{Re}\cdot\widetilde{v}''(z) + \beta_1(z)\cdot\widetilde{v}'(z) + [\beta_2(z) + i\omega]\cdot\widetilde{v}(z) + \alpha_1(z)\cdot\widetilde{r}_2'(z) + \alpha_2(z)\cdot\widetilde{r}_2(z) + \theta_1(z)\cdot\widetilde{r}_1'(z)$$

$$+\theta_2(z)\cdot\widetilde{r}_1(z) + \phi_1(z)\cdot\widetilde{t}'(z) + \phi_2(z)\cdot\widetilde{t}(z) = 0,$$

$$\beta_3(z)\cdot\widetilde{v}'(z) + \beta_4(z)\cdot\widetilde{v}(z) + \theta_3(z)\cdot\widetilde{r}_1'(z) + (\theta_4(z) - i\omega)\cdot\widetilde{r}_1(z) + \alpha_3(z)\cdot\widetilde{r}_2(z) = 0,$$

$$\beta_3(z)\cdot\widetilde{v}'(z) + \beta_5(z)\cdot\widetilde{v}(z) + \theta_3(z)\cdot\widetilde{r}_2'(z) + (\alpha_4(z) - i\omega)\cdot\widetilde{r}_2(z) + \theta_5(z)\cdot\widetilde{r}_1(z) = 0,$$

$$\frac{\lambda}{Pe}\cdot\widetilde{t}''(z) + \phi_3(z)\cdot\widetilde{t}'(z) + (\phi_4(z) + i\omega)\cdot\widetilde{t}(z) + \alpha_5(z)\cdot\widetilde{r}_2'(z) + \alpha_6(z)\cdot\widetilde{r}_2(z)$$

$$+\theta_6(z)\cdot\widetilde{r}_1'(z) + \theta_7(z)\cdot\widetilde{r}_1(z) + \beta_6(z)\cdot\widetilde{v}(z) = 0.$$

(8)

Further, using discretization by the finite-difference method (central approximation), a system of linear algebraic equations was obtained, which in matrix form has the following form:

$$(iN - \omega I)X = 0 \tag{9}$$

here $i$ is the imaginary unit; $X = (\widetilde{v}_k, \widetilde{r}_{1k}, \widetilde{r}_{2k}, \widetilde{t}_k)^T$ is the column vector of variable values at each step; $I$ is the identity matrix; $N$ is the matrix of coefficients for $X$ variables.

Note that the dimension of the above matrices depends on the number of nodes chosen during the discretizing of the system (8). We also should note that the estimated faulty proportion of the central differences applying is the smallest one as compared with the left and right approximations and it is proportional to the squared discretization interval.

According to Equation (9) it can be seen that $\omega$ is the eigenvalue of the coefficient matrix $N$. Since the natural frequency is a complex number, then $\omega = \omega_r + i\omega_i$, where $\omega_i$ is the increase coefficient.

Moreover, this coefficient makes it possible to judge whether oscillations are damped or growing. If all $\omega_i < 0$, then we can say that the oscillations are damped, which means that the state under study (stationary flow) is stable, otherwise, at $\omega_i > 0$ it is unstable [24]. Thus, the problem of stability analysis has been reduced to the problem of finding the eigenvalues of the coefficient's matrix. It should be noted that to determine the stability, it is sufficient to estimate only the maximum value of the imaginary part $\omega_i^{(1)}$, the so-called damping coefficient of the first mode.

## 3. Numerical Study of the Capillary Drawing Stability

*3.1. Isothermal Process*

### 3.1.1. Linear Stability

Based on the obtained stability model, a software package was developed. Depending on the input parameters of the process, it finds a stationary solution, forms a matrix of coefficients for a given number of discretization nodes, and determines the growth factor. The created complex allowed us to obtain a number of important theoretical and practical results. However, as mentioned earlier, one of the stages of the stability determination algorithm is the use of the finite difference method. Therefore, at first, the influence of the discretization step on the accuracy of calculations was evaluated and the optimal step was chosen.

To do this, at fixed system parameters and for a various number of partition points, the damping coefficients of the first modes $\omega_i^{(1)}$ were calculated. The observation was carried out for the relative deviation of the following type:

$$\Delta\omega_{i,j}^{(1)} = \frac{\omega_{i,j}^{(1)} - \omega_{i,j-100}^{(1)}}{\omega_{i,j-100}^{(1)}} \tag{10}$$

where $j = 100, 200, \ldots, 1000$ is the number of partitions in the numerical experiment.

Calculations show that when the number of discretization points is more than 500, the calculation result changes by less than 0.1%. All further calculations were done with this accuracy.

For the fiber manufacturing process, there is a set of technological parameters that can be adjusted in order to achieve the desired drawing results. Here is a brief overview of the parameters that are key to assessing the stability of the fiber manufacturing process. The most important of them is the drawing ratio $E$. The influence of this indicator was assessed and the influence of the geometrical parameters of the preform on the stability of the process were simultaneously considered (Figure 5).

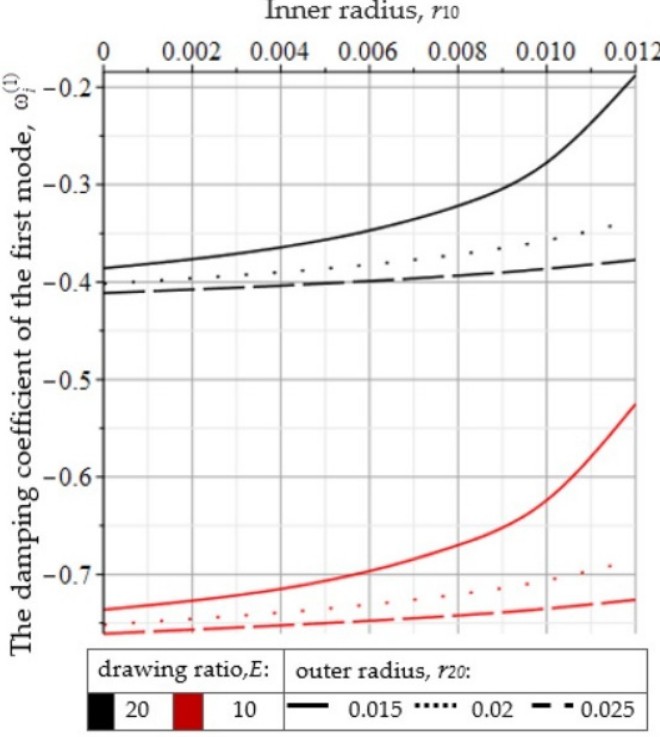

**Figure 5.** Dependence of $\omega_i^{(1)}$ on the geometry of the preform at various drawing ratios ($E$).

It can be said that with an increase in the drawing ratio, the process becomes less stable. In addition, reducing the thickness of the preform while increasing the size of the capillary cavity also leads to buckling. It should be noted that this dependence between the ratio and the stability of drawing was also revealed in the case of a solid fiber [27–30].

The main interest is the connection between the process stability and the Reynolds number, which characterizes the action of inertia forces and viscous friction. The drawing ratio was fixed at $E = 20$, and stability calculations were made at various values of the Reynolds number and for various geometric parameters of the preforms. In this case, the main parameters of the model were chosen according to the quartz fiber drawing technology [11,20,23]. The change in the Reynolds number was carried out by changing the viscosity values within the allowable values due to production. In this case, the Reynolds number took a value of no more than 1.5. In particular, this is consistent with the studies [28–30] on solid fibers, in which the range of the Reynolds numbers is the same.

Based on the results of calculating the coefficients of the first modes (Figure 6), it can be established that with an increase in the Reynolds number, the process stability of drawing a quartz capillary increases. In addition, an increase in the inner radius of the capillary or, what is the same, the radius of its cavity, leads to the stability loss of the drawing process.

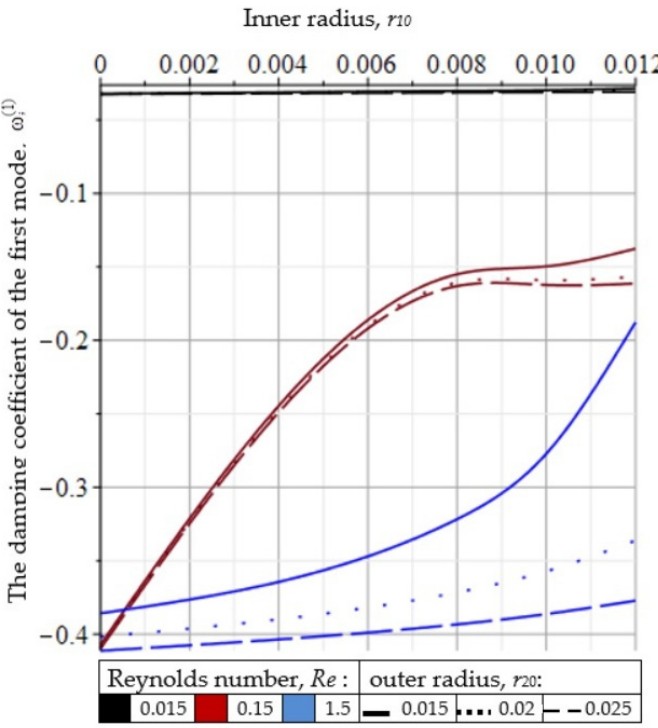

**Figure 6.** Dependence of $\omega_i^{(1)}$ on the preform radii at various *Re* numbers.

The analysis performed made it possible to find the stability limits to small perturbations for the drawing process, considering their geometric parameters, ratio, and Reynolds numbers. However, supercritical regions and secondary flows that develop as a result of the main flow stability loss remained unconsidered. The sensitivity level of the process to possible oscillations of the initial and boundary conditions of the system (7) could not be estimated. At zero initial and boundary conditions, the system (7) has a trivial solution, which means that the process fully corresponds to its stationary state. Small oscillations in the initial and/or boundary conditions result in deviations in the calculated solutions from the stationary ones. Therefore, it is obvious to note the possible growth of the existing perturbations and the stability loss of the drawing process. It is also of interest to compare the results of such a simulation with the results given by a nonlinear model calculated for the same oscillations. A numerical analysis of the complete system of nonlinear Equation (5)

with parameters from the region of linear stability and instability was made to answer the above questions.

### 3.1.2. Numerical Sensitivity Analysis

To run numerical experiments in the Comsol Multiphysics 5.2 simulation system, a software package was developed. This allows to solve the original nonlinear system describing the drawing process (5).

Modeling of the drawing process was made with parameters $Re = 15.2 \times 10^{-5}$, $Fr = 3.4 \times 10^{-5}$, $We = 0.22$, $La = 0$, $Ma = 144{,}662$ for a preform with radii $r_{10} = 0.01$, $r_{20} = 0.025$ at drawing ratio $E = 20$, which corresponds to a stable state.

The drawing process stability at the chosen values of the parameters means that the shapes of the profiles of the jet radius and the melt velocity approach the shapes corresponding to the stationary states. To analyze the sensitivity of the system, harmonic perturbations were introduced into the boundary conditions (5) for the values of the preform radius or the drawing speed at the inlet:

$$V(0, \tau) = \tfrac{1}{E} \cdot (1 + A \sin(2\pi\gamma\tau)), V(1, \tau) = 1 \cdot (1 + A \sin(2\pi\gamma\tau)),$$
$$R_2(0, \tau) = R_{20} \cdot (1 + A \sin(2\pi\gamma\tau)), R_1(0, \tau) = R_{10} \cdot (1 + A \sin(2\pi\gamma\tau)),$$

Since the most important indicator of the fibers drawing quality is the exact correspondence of the fiber proportions to the required dimensions (stationary), it is necessary to know what effects possible perturbations will have. Observation of the introduced perturbations was reduced to the control of the geometric characteristics of the finished fiber. The frequency effect of introduced oscillations on the stability of the capillary radius was evaluated. The calculations were carried out at a fixed amplitude $A = 0.05$, the frequency $\gamma$ varied in the range from 0 to 30.

The analysis performed showed that oscillations with frequency $\gamma = 25$ persist for the longest time, while the maximum amplitude at various frequencies is almost the same (Figure 7). In further sensitivity studies, the frequency of the introduced oscillations was taken as 25 in order to identify the worst possible scenario.

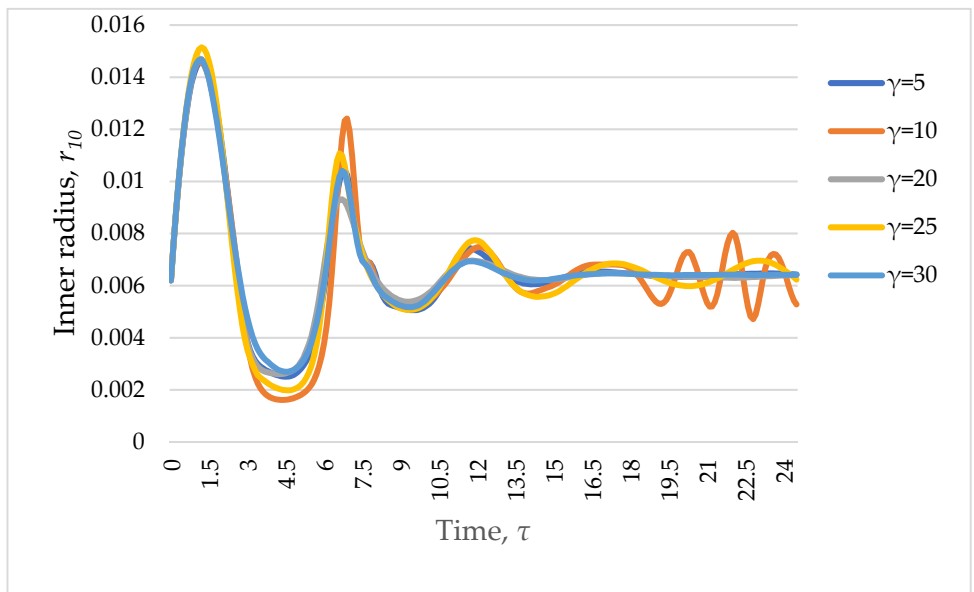

**Figure 7.** Oscillations in the values of the fiber radius in time (change in the frequency of harmonic oscillations of the initial conditions from 0 to 25).

Next, the influence of the amplitude A of the introduced oscillations on the geometry of the finished fiber was evaluated. As noted earlier, the perturbation frequency $\gamma = 25$ was fixed, and the amplitude A varied from 0.01 to 0.4. The calculation showed that the process

is not very sensitive to the amplitude of the introduced oscillations (Figure 8). Nevertheless, the highest sensitivity to oscillations with an amplitude of 0.05 was shown. Similarly to the previous one, the value $A = 0.05$ is fixed as the value of the perturbation amplitude, leading to the worst-case scenario in the study of stability.

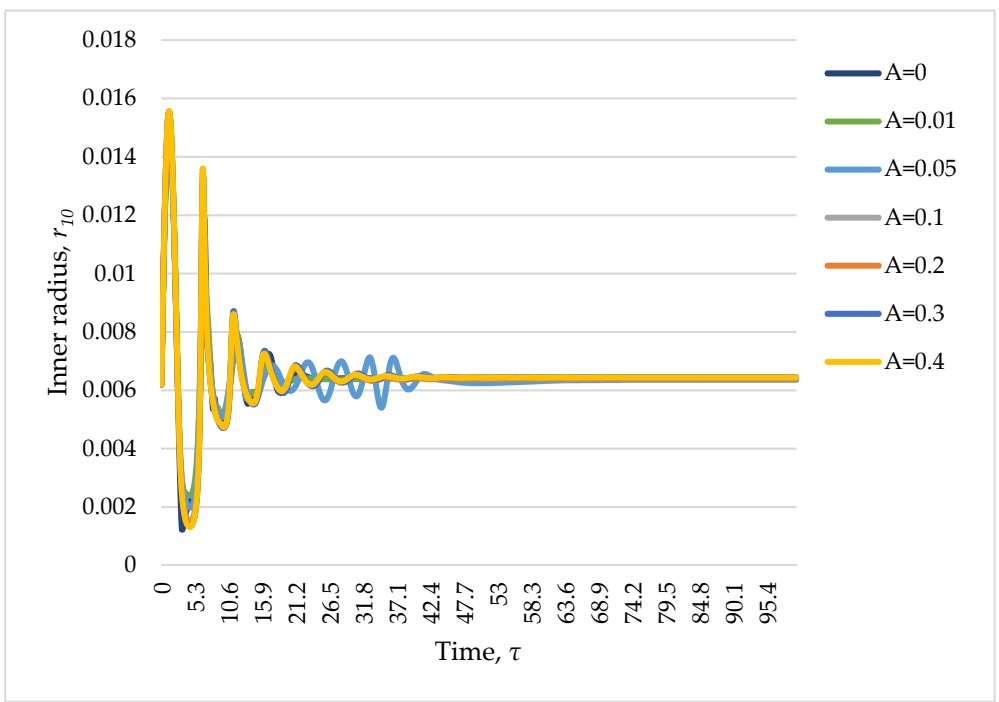

**Figure 8.** Oscillations in the values of the fiber radius in time (change in the amplitude of harmonic oscillations of the initial conditions).

So, the critical values of the amplitude and frequency of harmonic oscillations, leading to the longest process of establishing solutions, have been identified and fixed. Further, under the conditions of acting perturbations, the relative deviations of the outer and inner radii of the fiber from their stationary values $\Delta R_2(1, \tau)$ and $\Delta R_1(1, \tau)$ were calculated. Figure 9 shows the calculated values $\Delta R_1(1, \tau)$ as a function of time for perturbing actions specified at the boundaries of the computational region.

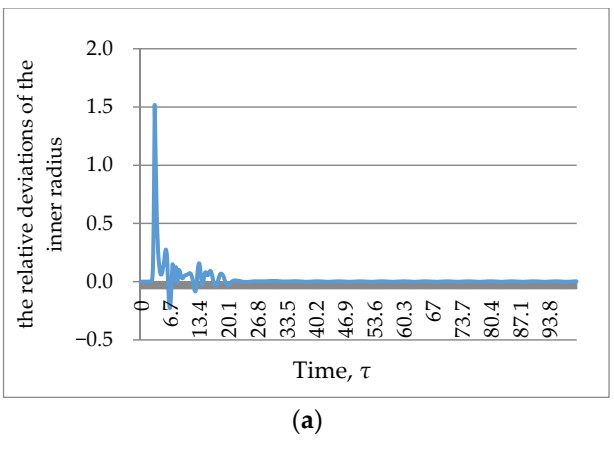

(a)

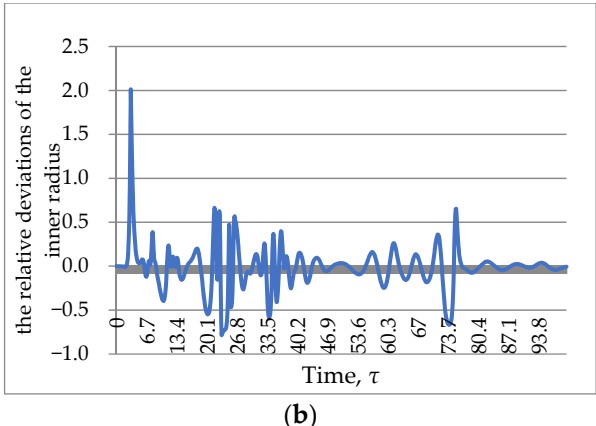

(b)

**Figure 9.** *Cont.*

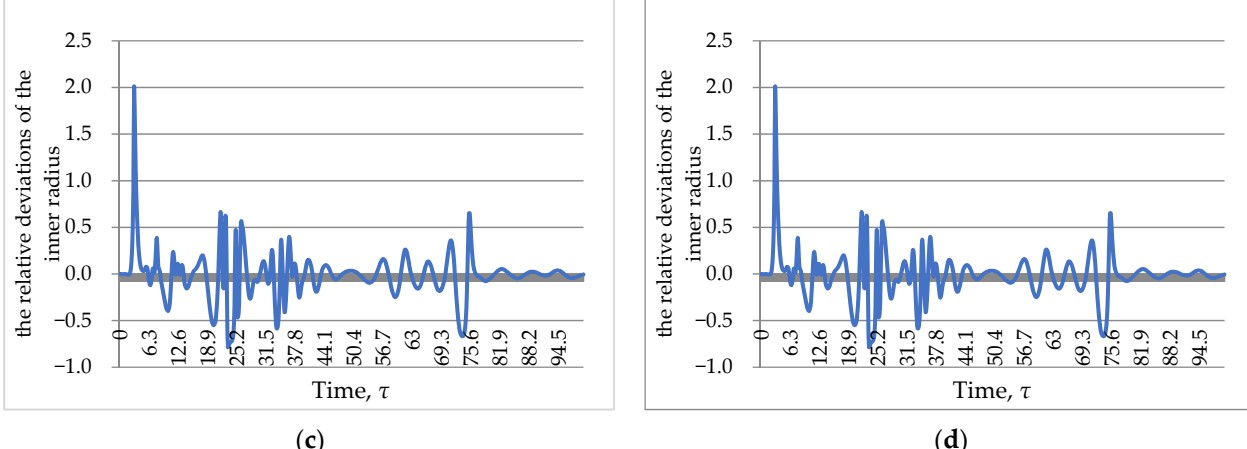

(**c**)                    (**d**)

**Figure 9.** Deviation of the fiber inner radius under various perturbing influences: (**a**) On the inner radius of the preform; (**b**) on the outer radius of the preform; (**c**) on the drawing speed of the finished fiber; (**d**) on the preform feed rate.

Similar results were also obtained in the calculation of $\Delta R_2(1, \tau)$. The form of the solutions presented in Figure 9 makes it possible to judge the damping of the perturbation amplitude with time. So, we note that when various types of boundary conditions are perturbed by 5%, the relative deviations $\Delta R_2(1, \tau)$, $\Delta R_1(1, \tau)$ change by no more than 1%.

Thus, the conducted studies confirmed that when choosing the parameters of the technological process from the stability region, small perturbations damp in a finite time.

Of no less interest is the behavior of the system in the zone of instability and the resulting secondary flows. As in the case of the steady mode, observations were made of the kinematic and geometrical parameters of the fiber. First, the behavior of the system was evaluated in the absence of perturbing influences. The calculation results for various values of the drawing ratio (instability region) are shown in Figure 10 as a dependence of the capillary wall thickness on time.

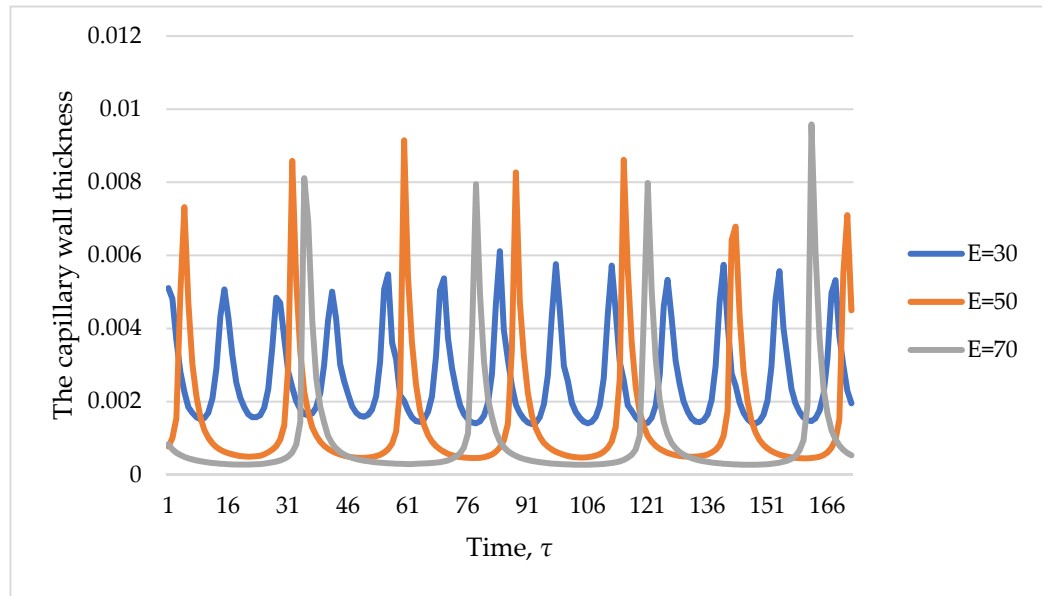

**Figure 10.** Time dependence of capillary wall thickness.

Numerical studies have shown that when the draw ratio is higher than the critical one (unstable mode), the process is not established with time. Oscillations, the amplitude of

which depends on the drawing ratio, remain. Moreover, the higher the ratio in relation to the critical value, the greater the amplitude of the resulting oscillations (Figure 11).

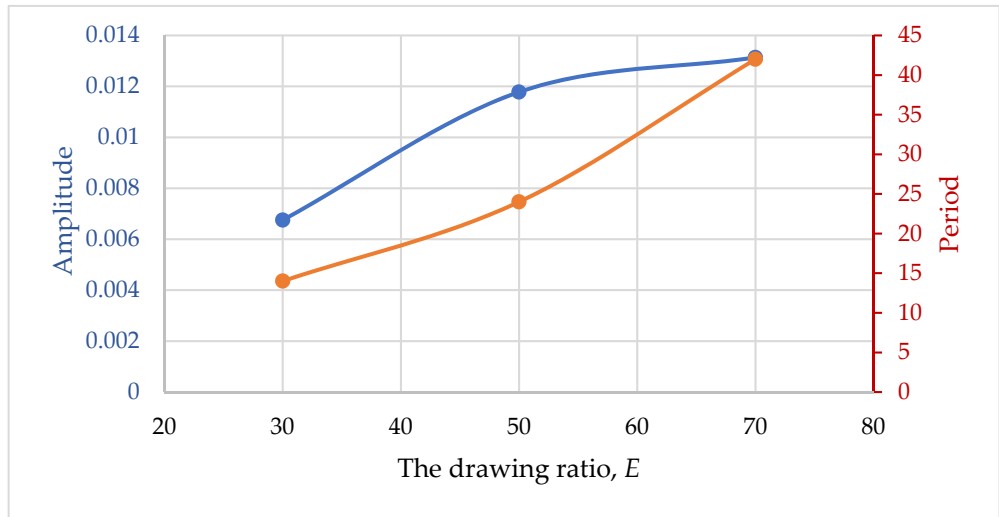

**Figure 11.** Dependence of the amplitude and period of oscillations on the drawing ratio.

We also note that the evolution of oscillations is stable (Figure 12). That is a mode of steady oscillations arises with a certain amplitude and period, the order of which depends on the drawing ratio.

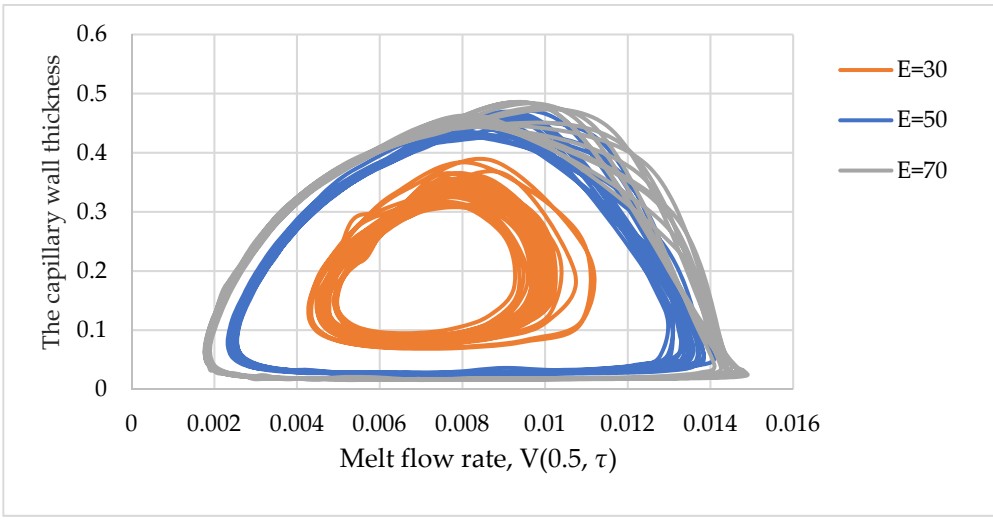

**Figure 12.** Dynamic system attractor.

*3.2. Non-Isothermal Process*

The numerical results presented above refer to the isothermal process of capillary drawing. From a practical point of view, it is more interesting to consider fiber drawing under non-isothermal conditions. Since, at a qualitative level, the processes in isothermal and non-isothermal flows are largely similar, in this study we will focus on the effect of temperature distribution along the furnace surface on the stability of drawing quartz tubes. It is known that the temperature along the surface of the furnace varies, and in the central part of the furnace, a zone (core) of width $H$ can be distinguished, in which the temperature is constant and much higher than near the edges [21].

Therefore, during the studies, the temperature distribution was set in the following form:

$$T_f(z,\tau) = \begin{cases} T_{f1}, z \in \left[0; \frac{(1-h)\cdot L}{2}\right] \\ T_{f2}, z \in \left(\frac{(1-h)\cdot L}{2}; \frac{(1+h)\cdot L}{2}\right) \\ T_{f1}, z \in \left[\frac{(1+h)\cdot L}{2}; L\right], \end{cases}$$

where $h = \dfrac{H}{L}$ is a relative width of the core of the heating element.

The temperature distribution along the surface of the heating element, along with the drawing ratio, preform feed rate and drawing speed, is a parameter that affects the stability of the fiber production process.

The aim of the study was to identify such values of the parameter $h$, at which the drawing process is stable. As before, all calculations were made in Comsol Multiphysics 5.2 with the drawing ratio from the stability region. The values of the remaining parameters of the drawing process were as follows:

$$v_0 = 0.01, \ r_f = 0.02, \ r_{10} = 0.008, \ r_{20} = 0.015, T_{f1} = 1600, \ T_{f2} = 2100$$

The study results presented in Figure 13 demonstrate a significant dependence of the drawing stability on the parameter $h$. The optimal zone $h \in [0.2; 0.7]$ in which the fiber formation is most stable is revealed. As in the isothermal case, the effect of ratio on stability is significant. Thus, the regularities obtained make it possible to design furnace temperature control systems in order to increase the stability of the drawing process.

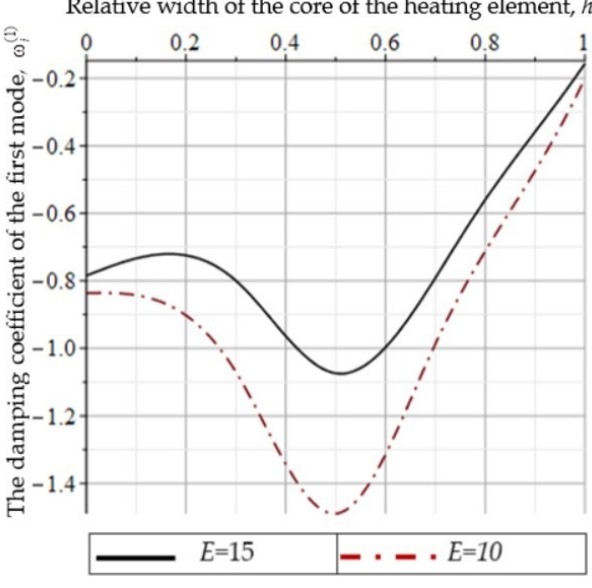

**Figure 13.** Dependence of $\omega_i^{(1)}$ on $h$ at various values of drawing ratio $E$.

A significant dependence of the drawing process stability on the furnace radius was also revealed. Figure 14 shows the dependence of the first modes on the indicated distance and the furnace core $h$. At a fixed preform radius, with an increase in the furnace radius $r_f$, the distance between the quartz and heater surfaces increases, which leads to a more stable drawing process.

A rather interesting dependence of the drawing stability on the temperature at the edges of the furnace $(T_{f1})$ has been obtained. At $T_{f1}$ below the melting temperature of quartz, the stability of the process is practically independent of $T_{f1}$. At $T_{f1}$ above the melting temperature, as $T_{f1}$ increases, the drawing process becomes less stable (Figure 15).

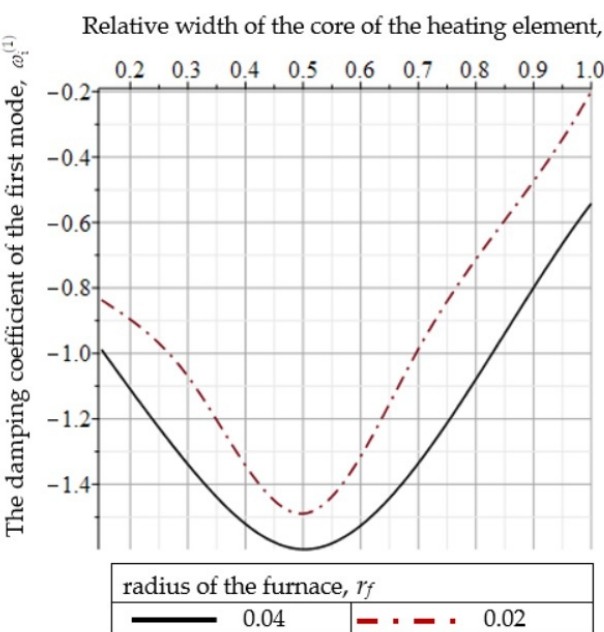

**Figure 14.** Dependence of $\omega_i^{(1)}$ on the radius of the furnace and *h*.

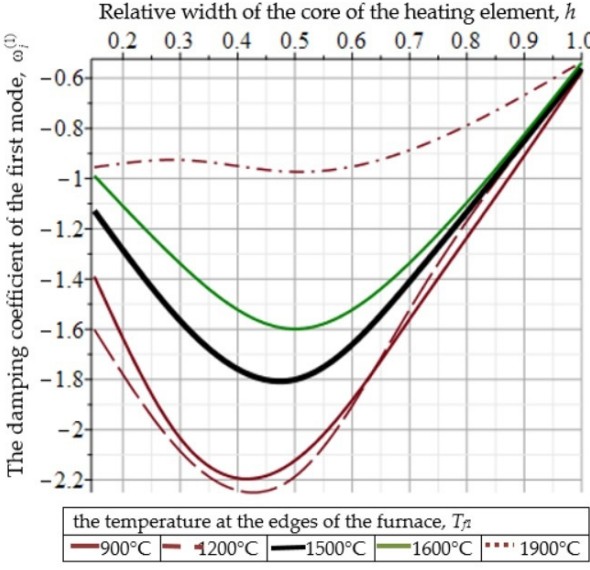

**Figure 15.** Dependence of $\omega_i^{(1)}$ on h for various values of $T_{f1}$.

Therefore, there are optimal parameters of the heating element: the core width *h*, its temperature $T_{f2}$, the temperature at the edges of the furnace $T_{f1}$ and the radius of the furnace $r_f$, at which the stability of the drawing process of quartz tubes increases significantly (several times).

## 4. Conclusions

During our research we studied the stability of the process of drawing quartz capillaries for photonic-crystal fibers. A modified PCF drawing model was obtained that considers the forces of inertia, viscous friction, and surface tension, as well as all types of heat transfer. Within the framework of the linear theory of stability, a mathematical model of capillary drawing was developed. The influence of the drawing ratio E and inertia forces (Reynolds number) on the stability of the process under consideration was estimated. At the same time, the influence of the preform geometric parameters did not go unnoticed.

For a complete nonlinear model of the isothermal drawing of a capillary, a numerical analysis of the sensitivity to perturbations arising in the boundary conditions was carried out. It was found that in the stability region constructed using the linear theory, perturbations of finite amplitude also fade over a finite period of time and do not significantly affect the quality of the finished product. As for the zone of linear instability, the model has a periodic solution, the amplitude and period of which depend on the drawing ratio.

The behavior of the secondary flow, which is formed as a result of the process transition to the instability region, is analyzed. It is shown that when the draw ratio is higher than the critical one (unstable mode), the process is not established with time, and the oscillation amplitude depends on the ratio E, and these oscillations are stable. The effect of the relative width of the heating element on stability was studied. It is shown that with an increase in the drawing ratio, the stability of the process is lost. Calculations for various values of the furnace radius showed that stability depends on the distance between the furnace and the preform surface: with the smaller radius, the stability of the process decreases. For the first time, the existence of optimal parameters of the heating element, such as the core width, its temperature, the temperature at the edges of the furnace, and the radius of the furnace, which guaranteed several times the drawing process stability increase, is shown.

The optimal heating zone, with the most stable fiber formation is revealed.

**Author Contributions:** Conceptualization, V.P.; methodology, V.P. and D.V.; software, A.D. and D.V.; validation, V.P., D.V. and A.D.; formal analysis, V.P.; investigation, A.D. and D.V.; resources, V.P.; data curation, V.P.; writing—original draft preparation, A.D.; writing—review and editing, D.V. and V.P.; visualization, A.D. and D.V.; supervision, V.P.; project administration, A.D. and V.P.; funding acquisition, V.P.All authors have read and agreed to the published version of the manuscript.

**Funding:** This research was carried out with the financial support of the Ministry of Science and Higher Education of the Russian Federation in the framework of the program of activities of the Perm Scientific and Educational Center "Rational Subsoil Use".

**Data Availability Statement:** Data and sources that are not openly available may be provided by the corresponding author on reasonable request.

**Conflicts of Interest:** The authors declare no conflict of interest.

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
