# Peer review of "Mathematical Modeling of Capillary Drawing Stability for Hollow Optical Fibers"

_algorithms, doi:10.3390/a16020083_

Round 1
Reviewer 1 Report
The authors studied the stability of the process of drawing quartz capillaries for photonic-crystal fibers. They obtained a modified PCF drawing model that considers the forces of inertia, viscous friction, surface tension, and heat transfer. They developed a mathematical model of capillary drawing within the framework of the linear theory of stability. The work is impressive and I recommend accepting and publishing the manuscript.
Author Response
The authors express their sincere gratitude to Reviewer for attention to the work and valuable comments.
Reviewer 2 Report
The authors solved the stability problems of the drawing process using the modified capillary drawing model (based on linear theory), which considers inertial, viscous and surface tension forces, as well as all types of heat transfer. They also developed algorithms for the numerical solution of the problem.
Comments:
1. The literature review is inadequate.
2. Give the assumptions for the present mathematical model (PDEs).
3. The drawing of Figure 2, 3 15 & 16 is not clear. Redraw it clearly.
4. Why do the central difference approximations (finite difference method) only used to using discretization?
5. What is meant by “all types of heat transfer”? is the model applicable to all heat transfer process, like, conduction, convection, radiation and mixed of these?
6. Why the values of Reynolds number used is too small? It is better to take upto 1000.
7. Please Add the version of Comsol Multiphysics
8. What are the limitations of the model and study.
Author Response
Point 1: The literature review is inadequate.
Response 1: The introduction was rewritten. Besides, the benchmarking was added too and the literature review was rather expanded. As a result, the list of references has increased, and the numbering of references has also changed.
Point 2: Give the assumptions for the present mathematical model (PDEs).
Response 2: he chapter “Mathematical model of drawing quartz capillaries” was supplemented by the limitations of the obtained model.
Point 3: The drawing of Figure 2, 3 15 & 16 is not clear. Redraw it clearly..
Response 3: Figures have been corrected by clarity improving
Point 4: Why do the central difference approximations (finite difference method) only used to using discretization?
Response 4: We added an explanation of the central difference use for solving the system on page 10.
Point 5: What is meant by “all types of heat transfer”? is the model applicable to all heat transfer process, like, conduction, convection, radiation and mixed of these?
Response 5: The meaning of the phrase "All types of heat transfer" are described in more detail in the chapter “Mathematical model of drawing quartz capillaries” on page 4
Point 6: Why the values of Reynolds number used is too small? It is better to take upto 1000
Response 6: In the chapter "Numerical Modeling" it was additionally explained what kind of values for the Reynolds number are taken and why.
Point 7: Please Add the version of Comsol Multiphysics.
Response 7: . In the article where Comsol Multiphisics is mentioned (p. 6, 13), a version of this software product was added, namely Comsol Multiphisics 5.2.
Point 8: What are the limitations of the model and study.
Response 8: We tried to answer this question in the introduction and we provided a little bit more details in the chapter “Mathematical model of drawing quartz capillaries” of model limitations.
The authors express their sincere gratitude to Reviewer for attention to the work and valuable comments.
Reviewer 3 Report
Please see the attached file

Author Response
Point 1: Authors should include major outcomes in the abstract.
Response 1: The abstract was finalized according to the comment.
Point 2:.I suggest to strength the novelty of this study. What do you do different from the other studies? What is the limit of previous studies?
Response 2:The introduction was rewritten with taking into account this comment.
Point 3: A benchmarking should be added to your work. It means that you should evaluate your results by comparison with other published papers in the field.
Response 3: The benchmarking was added and the literature review was rather expanded. As a result, the list of references has increased, and the numbering of references has also changed.
Point 4: Add suitable references for governing Eqs.
Response 4: In the chapter “Mathematical model of drawing quartz capillaries" corresponding links were added for the control equations (page 5).
Point 5: All presented equations should be written in Math Type's Equation, and all parameters should be in italics in the entire manuscript.
Response 5: Mathematical formulas were designed according to the requirements of the journal and equetions as well as parameters were corrected for the entire manuscript.
Point 6: Potentials and Limitations and how to overcome these limitations should be presented.
Response 6: We tried to answer this question in the introduction and we provided a little bit more details in the chapter “Mathematical model of drawing quartz capillaries” of model limitations.
Point 7: Please improve the quality of all figures.
Response 7: Figures have been corrected by clarity improving.
Point 8: Please ensure that all references listed at the end of the paper should be written according to the formatting guidelines in the journal.
Response 8: The list of references and mathematical formulas were designed according to the requirements of the journal.
Point 9: Finally, it is recommended that the wordings and grammar of English should be rechecked throughout the present manuscript.
Response 9: English grammar has been edited throughout the manuscript.
The authors express their sincere gratitude to Reviewer for attention to the work and valuable comments.
Round 2
Reviewer 2 Report
The authors revised the manuscript well and the explanations are given in detail. The paper is suitable for publication.
Reviewer 3 Report
I have seen the efforts by the authors to improve the paper. It can be accepted for publication in the present form.